# Investigation on Human Serum Protein Depositions Inside Polyvinylidene Fluoride-Based Dialysis Membrane Layers Using Synchrotron Radiation Micro-Computed Tomography (SR-μCT)

**DOI:** 10.3390/membranes13010117

**Published:** 2023-01-16

**Authors:** Amira Abdelrasoul, Ning Zhu, Ahmed Shoker

**Affiliations:** 1Department of Chemical and Biological Engineering, University of Saskatchewan, 57 Campus Drive, Saskatoon, SK S7N 5A9, Canada; 2Division of Biomedical Engineering, University of Saskatchewan, 57 Campus Drive, Saskatoon, SK S7N 5A9, Canada; 3Canadian Light Source, 44 Innovation Blvd, Saskatoon, SK S7N 2V3, Canada; 4Nephrology Division, College of Medicine, University of Saskatchewan, 107 Wiggins Rd, Saskatoon, SK S7N 5E5, Canada; 5Saskatchewan Transplant Program, St. Paul’s Hospital, 1702 20th Street West, Saskatoon, SK S7M 0Z9, Canada

**Keywords:** hemodialysis membrane, PVDF, protein adsorption, albumin, fibrinogen, transferrin

## Abstract

Hemodialysis (HD) membrane fouling with human serum proteins is a highly undesirable process that results in blood activations with further severe consequences for HD patients. Polyvinylidene fluoride (PVDF) membranes possess a great extent of protein adsorption due to hydrophobic interaction between the membrane surface and non-polar regions of proteins. In this study, a PVDF membrane was modified with a zwitterionic (ZW) polymeric structure based on a poly (maleic anhydride-alt-1-decene), 3-(dimethylamino)-1-propylamine derivative and 1,3-propanesultone. Fourier transform infrared spectroscopy (FTIR), scanning electron microscope (SEM), and zeta potential analyses were used to determine the membrane’s characteristics. Membrane fouling with human serum proteins (human serum albumin (HSA), fibrinogen (FB), and transferrin (TRF)) was investigated with synchrotron radiation micro-computed tomography (SR-μCT), which allowed us to trace the protein location layer by layer inside the membrane. Both membranes (PVDF and modified PVDF) were detected to possess the preferred FB adsorption due to the Vroman effect, resulting in an increase in FB content in the adsorbed protein compared to FB content in the protein mixture solution. Moreover, FB was shown to only replace HSA, and no significant role of TRF in the Vroman effect was detected; i.e., TRF content was nearly the same both in the adsorbed protein layer and in the protein mixture solution. Surface modification of the PVDF membrane resulted in increased FB adsorption from both the protein mixture and the FB single solution, which is supposed to be due to the presence of an uncompensated negative charge that is located at the COOH group in the ZW polymer.

## 1. Introduction

The development of polymer chemistry has offered a broad variety of materials for developing hemodialysis (HD) membrane technology. The very first HD membrane was made out of cellophane [1]. During the last 40 years of HD development, different polymers, such as poly(methyl methacrylate) (PMMA), polyacrylonitrile (PAN), polyethylenimine (PEI), polyvinylidene fluoride (PVDF), polyamide (PA), cellulose triacetate (CTA), and poly(vinyl alcohol) (PVA), have been used as the main material for membrane fabrication [2,3,4,5,6,7,8,9]. Currently, the polysulfone family of polymers (polysulfone (PES), polyarylethersulfones (PAES), etc.) are used for the production of high-performance membranes, not only for HD but for other biomedical applications too, due to their excellent balance of thermal and chemical resistance and mechanical properties [10,11,12,13].

However, interest in other polymers still exists because of the high price of polysulfone polymers. The limitations of other polymers are determined by their surface chemistry. PVDF possesses high hydrophobicity, resulting in it being prone to protein adsorption and membrane–surface interactions, which are considered highly undesirable phenomena for HD. One of the ways to overcome this disadvantage is to perform chemical modification and alter membrane surface chemistry [14,15,16,17,18]. The main idea of this modification is to create a hydrophilic layer on the membrane surface that prevents protein adsorption via hydrophobic interaction.

This approach was implemented in three generations of HD membranes [19]. The first generation utilized the presence of OH groups, which possess high hydrophilic properties [20,21]. However, high hydrophilicity does not necessarily mean high hemocompatibility, and OH-containing polymers often trigger complement cascade reactions, resulting in severe health problems for HD patients. The second generation of HD membranes appeared from surface modification with ethylene glycol-based polymers [20,21]. The less hydrophilic O-containing surface was believed to trigger fewer blood coagulation processes and to reduce fibrinogen adsorption and the coagulation process [18,22].

The latest (third) generation of HD membranes are covered with zwitterionic (ZW) structures, which comprise equal number of anionic and cationic moieties, resulting in a total zero charge. Despite being uncharged, the ZW molecule is able to strongly bind water molecules, thus creating electrostatically induced hydration that is believed to hinder protein adsorption on the membrane surface [19,23,24,25,26,27,28]. This approach has been successfully implemented for many polymer structures, including PVDF membranes, resulting in a significant improvement in their biocompatibility and antifouling properties [29,30,31,32].

Recently, our research group has reported the challenges, trends, and current state of ZW-modified membranes and has also conducted research showing the improvement in antifouling properties via surface modification with zwitterion coatings [33,34,35,36,37,38]. Our group has studied the interaction of human blood proteins with membrane models using molecular docking. Those studies were crucial to providing insights and highlighting the functional group(s) that are responsible for the interactions with human serum proteins. Based on the computational results, our research group has developed novel ZW membranes that reduced the inflammatory biomarker released in dialysis patients’ serum [35,36,37,38].

The aim of the current paper is to investigate human serum protein depositions inside polyvinylidene fluoride-based dialysis membrane layers using synchrotron radiation micro-computed tomography (SR-μCT). We also study the influence of the ZW structure on PVDF membrane fouling with proteins.

## 2. Materials and Methods

### 2.1. Materials

PVDF membranes as sheet material were supplied from Sterlitech. Human serum protein (human serum albumin (HSA), fibrinogen (FB), and transferrin (TRF)) were purchased from Sigma-Aldrich, Oakville, Canada. Gold nanoparticles were purchased from Nanopartz™, Loveland CO, USA. These nanoparticles were conjugated to human proteins (albumin, fibrinogen, and transferrin) to be visualized in the SR-μCT. Phosphate buffer solution, ethanol (≥99.9%), chloroform (≥99.5%), methanol (≥99.9%), 1,3-propanesultone, and poly (maleic anhydride-alt-1-decene), 3-(dimethylamino)-1-propylamine derivative (PMAL^®^-C8, BioReagent, for molecular biology) were purchased from Sigma-Aldrich, Oakville, Canada. Saline solution was provided by St. Paul’s Hospital.

### 2.2. Membrane Surface Modification

The procedure of PVDF membrane surface modification is described in our previous paper [38]. The bare PVDF membrane was vacuum-dried before being introduced into a 2:1:2 ratio suspension consisting of a premixed 90:10 chloroform–ethanol mixture and 50% PMAL^®^-C8. The membrane was completely immersed within this mixture and stirred (100 rpm at 25 °C). Approximately 25 μL of the synthesized polymer solution was utilized in modifying the membrane. The membrane was then removed from the polymer solution and washed in ethanol (50%) for 10 min and ultrapure deionized water for 10 min in order to remove the adhering polymer solution. Subsequently, the membrane was completely immersed in a propanesultone and methanol mixture at 40 °C for 60 min in order to complete the ring-opening polymerization reaction with PMAL^®^-C8. Finally, ethanol-washing and vacuum-drying procedures were carried out before storage. The chemical structure of the resultant membrane is given in Figure 1. The modified membrane is referred to here and subsequently as the PVDF-ZW membrane.

### 2.3. Human Serum Protein Solution Preparation

Human serum protein solutions were prepared using a saline and phosphate buffer solution with resultant pH = 7.4. The concentrations of individual proteins and in their mixtures were as follows: HSA, 50 mg/mL; FB, 2 mg/mL; and TRF, 3 mg/mL. Each protein was conjugated with corresponding gold NPs before being mixed. The resultant solutions were injected into PVDF and PVDF-ZW membranes, with further analysis of the membranes using the X-ray based synchrotron technique.

### 2.4. In Situ Synchrotron Advanced Imaging Techniques at BioMedical Imaging and Therapy (BMIT) Beamline

Visualization of protein adsorption was conducted using a monochromatic beam at 20 keV energy. A beam monitor, AA-40 (500 μm LuAG scintillator, Hamamatsu, Japan), coupled with a high-resolution camera, PCO Dimax HS (PCO, Germany), providing a pixel size of 5.5 μm and a field of view (FOV) of 4.4 mm × 2.2 mm, was used to record the X-ray radiographs. A high photon flux allows for very detailed observation of particle deposition in microscopic layers of the membrane. CT projections were recorded at 20 keV at the 05ID-2 beamline of the BioMedical Imaging and Therapy (BMIT), at the Canadian Light Source (CLS), as presented in Figure 2a. The resulting radiographs were converted into graphical images using Avizo software. Further image analysis was performed using imageJ software, National Institutes of Health, BSD2. Gold nanoparticles conjugated with proteins produced the brightest spots on the image, thus providing quantitative information about the protein amount adsorbed at each scanned layer (see Figure 2b). In case of adsorption from multiprotein solutions, each protein was detected and analyzed on the basis of the specific shape of nanoparticles used for conjugation with each protein. Thus, spherical particles were used for conjugation with HSA, rods (sphericity ratio 0.85) were used for conjugation with FB, and cylinders (sphericity ratio 0.91) were used for conjugation with TRF. Detailed information about membrane imaging and radiograph representation with membrane division into layers and regions is described in our recent papers [38,39,40,41,42]. Membrane thickness was modeled with 10 regions of interest (ROIs), as shown in Figure 2c. Region 1 represents the very top membrane surface. The middle layers represent the internal membrane structure. The bottom membrane parts are located in Region 10. In order to ensure the accuracy of the data, four measurements were carried out for each sample. The data presented in the discussion are an average of the measurements.

### 2.5. Analytical Techniques Used for Membrane Characterization 

#### 2.5.1. Fourier Transform Infrared Spectroscopy ATR-FTIR Analyses

Prior to characterization, these membrane specimens were appropriately vacuum-dried for one day at 30 °C. They were then placed on a Renishaw-inVia Raman Microscope (Renishaw, UK) stage and were ready for surface chemical analysis using attenuated total reflectance Fourier transform infrared spectroscopy (ATR-FTIR). Both bare and ZW-coated PVDF membranes were probed for membrane-bound chemical groups with IR spectra, all recorded in transmittance mode.

#### 2.5.2. Scanning Electron Microscope (SEM)

The comparative surface morphologies of both ZW-coated and bare membranes were recorded with the aid of a Hitachi SU8010 Field-Emission Scanning Electron Microscope, Hitachi, Ibaraki, Japan. All SEM images were recorded at 3 kV accelerating voltage in order to avoid burning of the Cr-coated membrane samples (Q150T ES).

#### 2.5.3. Zeta Potential Analyzer

A Zetasizer-Nano (Malvern Instruments Ltd., Malvern, UK) was used to measure the HD membranes surface charge (zeta potential) with a precision of ±0.01 mV. The membrane sample preparation and software steps are detailed in our recent study [43].

## 3. Results and Discussion

### 3.1. Zwitterionic-Polymer-Modified and Unmodified PVDF Membrane Characteristics

Figure 3 shows the ATR-FTIR spectra of both zwitterionic-polymer-modified (PVDF-ZW) and unmodified (PVDF) membranes. As expected, the spectrum of the bare membrane shows peaks corresponding to C-F vibration, 1410 (bending) and 1199 (stretching) cm^−1^, as well as C-H vibrations at 2978 (stretching) and 1383 (deformation) cm^−1^ [43]. The alteration in PVDF peak intensities and positions for the PVDF-ZW sample attests to the surface polymer adsorption, especially the suppressed peaks at 2978 cm^−1^ (C-F bending) and 1383 cm^−1^ (C-F stretching) [44,45,46]. The PVDF membrane was polymerized with the zwitterionic polymer, and the effect of this process could be identified by ATR-FTIR analysis. Evidence of polymer adsorption on PVDF for the PVDF-ZW membrane can be seen in the absorption peaks related to stretching vibrations of N-H, C-N, and C-C around 3000, 1471, and 1000 cm^−1^, respectively. There are still characteristic absorption bands for N-H wagging at 764 cm^−1^. These are the main chemical groups of PMAL-C8 moiety on the zwitterionic polymer. It is important to note that these peaks are completely absent from the spectrum of the unmodified membrane [47]. The ring-opening polymerization reaction of 1,3-propanesultone after reaction with the tertiary amine on PMAL^®^-C8 yields new IR vibration peaks at 1619 cm^−1^ corresponding to N-H (bending). This marks the acylation of propanesultone ring in the presence of methanol. The presence of sulfonate $SO_{3}^{-}$(stretching) chemical groups as well as that of CH_3_ and CH_2_ (bending) could also be seen at 1039, 1381, and 1462 cm^−1^, respectively [47]. The symmetric and asymmetric stretching sulfonate vibrations that would have been seen at 1055 and 1238 cm^−1^, respectively, are not observed, which is due to the overlapping CF_2_ peak around 1182 cm^−1^. The band at 1664 cm^−1^ could be assigned to amide C=O stretching, whereas the one at 1524 cm^−1^ is N-H stretching of the O=C-NH groups in the polymer chain [48]. Compared with virgin PVDF, a new peak at 953 cm^−1^ appeared in the zwitterionic sulfobetaine PVDF membrane, which can be attributed to the quaternary amine groups. The observed evidence of ZW on the PVDF membrane shows that the chemical modification process via dip coating was successful. This indicates that the polymer coated the PVDF membrane. Tiny shoulder peaks consistent with the C-H bond are common for both membranes at 2900 cm^−1^.

Figure 4 depicts the surface morphologies of both bare and ZW-coated PVDF membranes. The presence of homogeneous fine strains, characteristic of the PVDF membrane, is conspicuous in Figure 4a. The magnified image in Figure 4b shows distinct microporous structures with different sizes (approximately 0.1 μm max). It is clear that the morphology of the pristine PVDF is quite different from that of the modified membrane, as can be seen in Figure 4b,d. The zwitterionic polymer significantly altered the PVDF membrane but did not block micropores, with no observable agglomeration. The membrane surface charge measurements showed evidence of zwitterionation of the PVDF membrane. The PVDF and PVDF-ZW have a surface charge of −2.5 and 0.35, respectively. 

### 3.2. Human Serum Protein Adsorption from Protein Mixture Solution

Based on the SR-μCT analysis, we detected a difference in the protein composition in the adsorbed layer on the membrane surface in comparison with the protein composition in the initial solution (see Figure 5). Thus, the content of HSA in the protein mixture in the initial solution is 90%, whereas this value drops to less than 80% in the protein mixture layer adsorbed by membranes. Moreover, the HSA content for the PVDF membrane is 4% higher than that for the PVDF-ZW membrane. Interestingly, the HSA content tends to decrease with membrane thickness (see Figure 6a) for both membranes.

It is to be noted that the change in HSA content is compensated by the corresponding increase (by more than 12%) in FB content predominately; i.e., no significant change in TRF content is observed, and its content remains on the same level for both the initial solution and the adsorbed protein layer. Notably, the FB content for the PVDF-ZW membrane is the highest (3% higher than that for PVDF membranes), indicating a higher interaction of the PVDF-ZW membrane with this protein. As well as for overall FB content, FB content inside the membrane also tends to increase (see Figure 6b), compensating the corresponding decrease in HSA content (see Figure 6a).

TRF content across membrane thickness possesses the same tendency to increase as in the case of FB, although the absolute change in TRF content is less than 1%, whereas FB content changes within 3–4%.

The protein–protein and protein–membrane interactions induced a gain of free energy, so the protein molecule underwent conformational changes and spread on the membrane surface as a part of the third stage. Depending on the degree of interaction, a large protein can dislocate a small one in a phenomenon called the Vroman effect. This dynamic interaction and equilibrium lead to progressive changes in the composition of the protein cake layer, affecting the filtration performance and biocompatibility profile of the surface [49,50,51,52,53].

Considering adsorbed protein distribution (see Figure 7), not quite uniform distribution could be observed for both membranes. Proteins were shown to tend to locate more at the bottom of the membrane (high index regions), whereas both membranes have a similar pattern in protein distribution. 

### 3.3. FB Adsorption from Its Single Solution

Besides the adsorption of the protein mixture, FB adsorption from its single solution was also studied, since this protein is believed to play a major role in the blood clotting process. Figure 8 demonstrates the difference in FB adsorption ratios when FB is adsorbed from its single solution and from a protein mixture. For both membranes, an increased FB adsorption ratio was observed when FB was adsorbed from the protein mixture, which can be explained by the Vroman effect. Thus, HSA has a concentration of 50 mg/mL, which results in its adsorption on the membrane surface and its further replacement with FB. Without HSA, the adsorption rate of FB seems to be reduced, resulting in a lower adsorption ratio. Moreover, the PVDF-ZW membrane possesses a slightly higher FB adsorption, which correlates with increased FB content in the adsorbed protein layer (see Figure 5).

The possible explanation for the increased FB adsorption by the PVDF-ZW membrane could be related to ZW polymer structure and charge distribution (see Figure 9). In addition, FB illustrated an anisotropic charge distribution of fibrinogen by computing the local charges of each domain [54,55]. Thus, it can be stated that this polymer contains an uncompensated negative charge located at the COOH groups. Although the COOH group possess weak acidic properties, this seems to be enough to affect protein adsorption, causing some increase in electrostatic interaction between the membrane surface and positive moieties in the FB molecule.

When FB distribution across membrane thickness is considered (see Figure 10), it can be observed that PVDF and PVDF-ZW membranes possess different behaviors. Thus, FB is uniformly distributed across PVDF-ZW membrane thickness, whereas unmodified PVDF holds FB predominately in the middle regions.

## 4. Conclusions

PVDF and PVDF-ZW membranes possess similar behaviors in protein adsorption. Thus, both membranes tend to adsorb more FB, resulting in a change in protein composition in the adsorbed layer in comparison with the protein mixture solution; i.e., FB content in the adsorbed protein layer increased from 3.6% (protein mixture solution) to 15% for PVDF and 18% for PVDF-ZW membrane. This increase in FB content was compensated by the corresponding decrease in HSA content, indicating the major role of HSA in the Vroman effect. At the same time, TRF content was not affected and remained approximately the same for the protein mixture solution and adsorbed protein layer for both membranes. This assumption was confirmed by comparing the FB adsorption ratio of the FB single solution and FB adsorption from the protein mixture solution. FB adsorption from the protein mixture solution was higher for both membranes. Notably, the PVDF-ZW membrane adsorbed more FB compared with the unmodified membrane. This difference is supposedly due to the structure of the ZW polymer used for modification. The presence of the uncompensated charge located on COOH groups resulted in a stronger interaction with the FB molecule, causing increased FB adsorption.

## Figures and Tables

**Figure 1 membranes-13-00117-f001:**
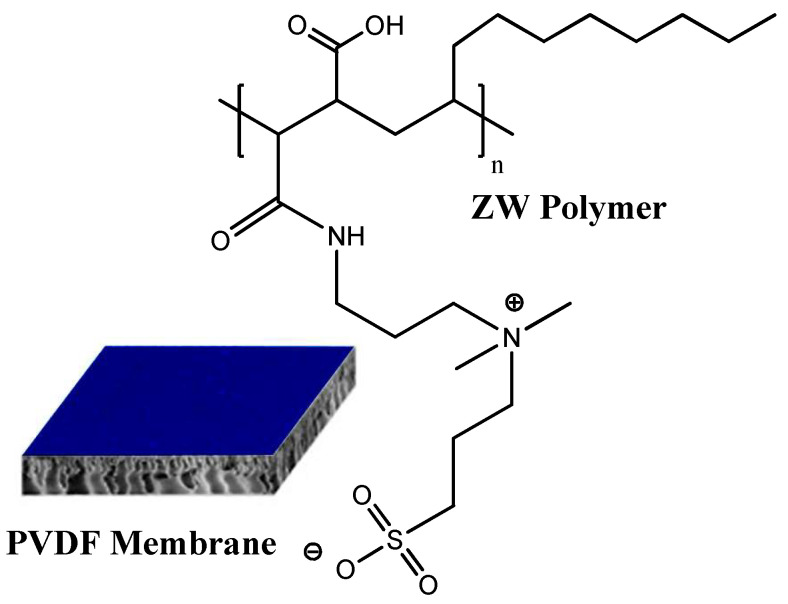
The structure of PVDF membrane modified with ZW polymer coating.

**Figure 2 membranes-13-00117-f002:**
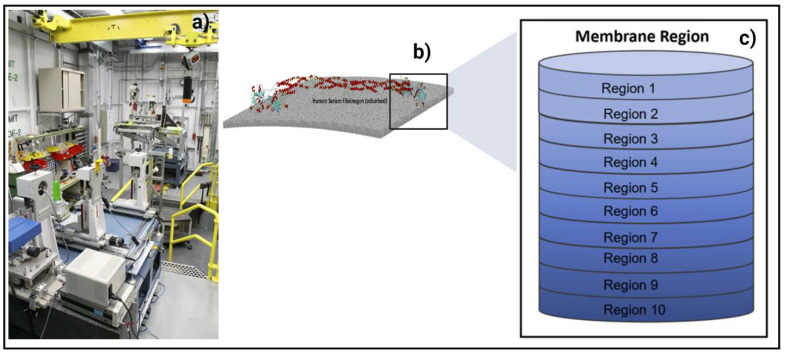
(**a**) Photo of the experimental SR-μCT system in BMIT hatch at the Canadian Light Source (CLS); (**b**) side view of membrane; and (**c**) layer-by-layer division of membranes in 10 regions.

**Figure 3 membranes-13-00117-f003:**
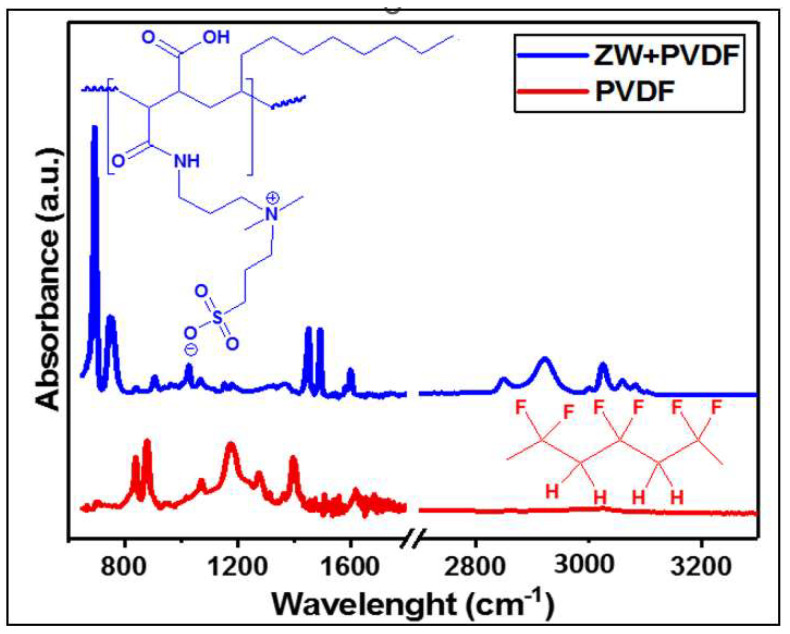
ATR-FTIR of unmodified PVDF and zwitterionic-polymer-modified PVDF membranes.

**Figure 4 membranes-13-00117-f004:**
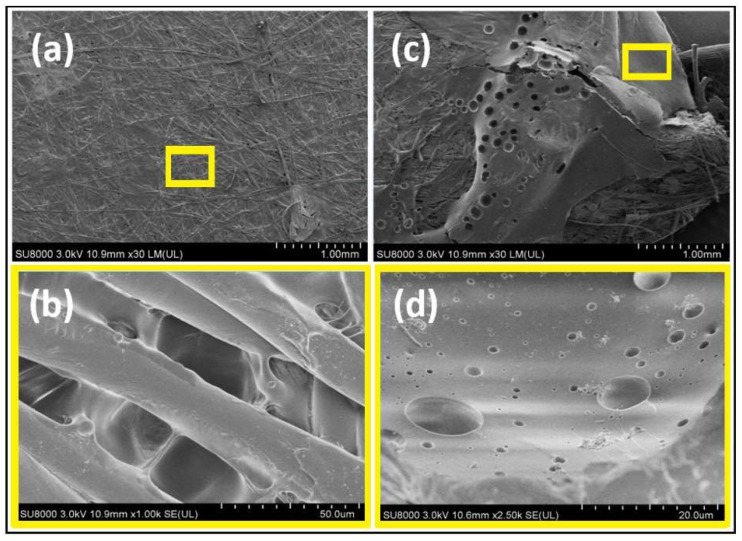
SEM micrographs of pristine PVDF and zwitterionic-polymer-modified PVDF membranes. (**a**) pristine PVDF membranes (**b**) microporous structures of pristine PVDF membranes (**c**) zwitterionic-polymer-modified PVDF membranes (**d**) microporous structures of zwitterionic-polymer-modified PVDF membranes.

**Figure 5 membranes-13-00117-f005:**
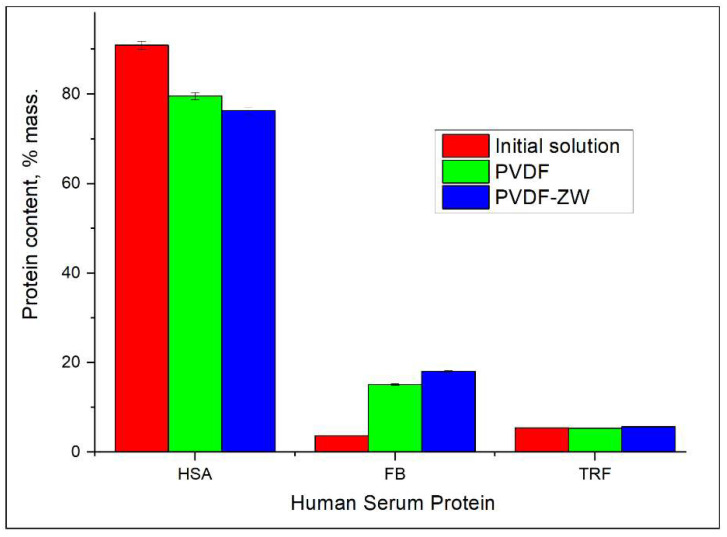
Comparison between protein composition in initial solution and in proteins adsorbed on the membrane surface.

**Figure 6 membranes-13-00117-f006:**
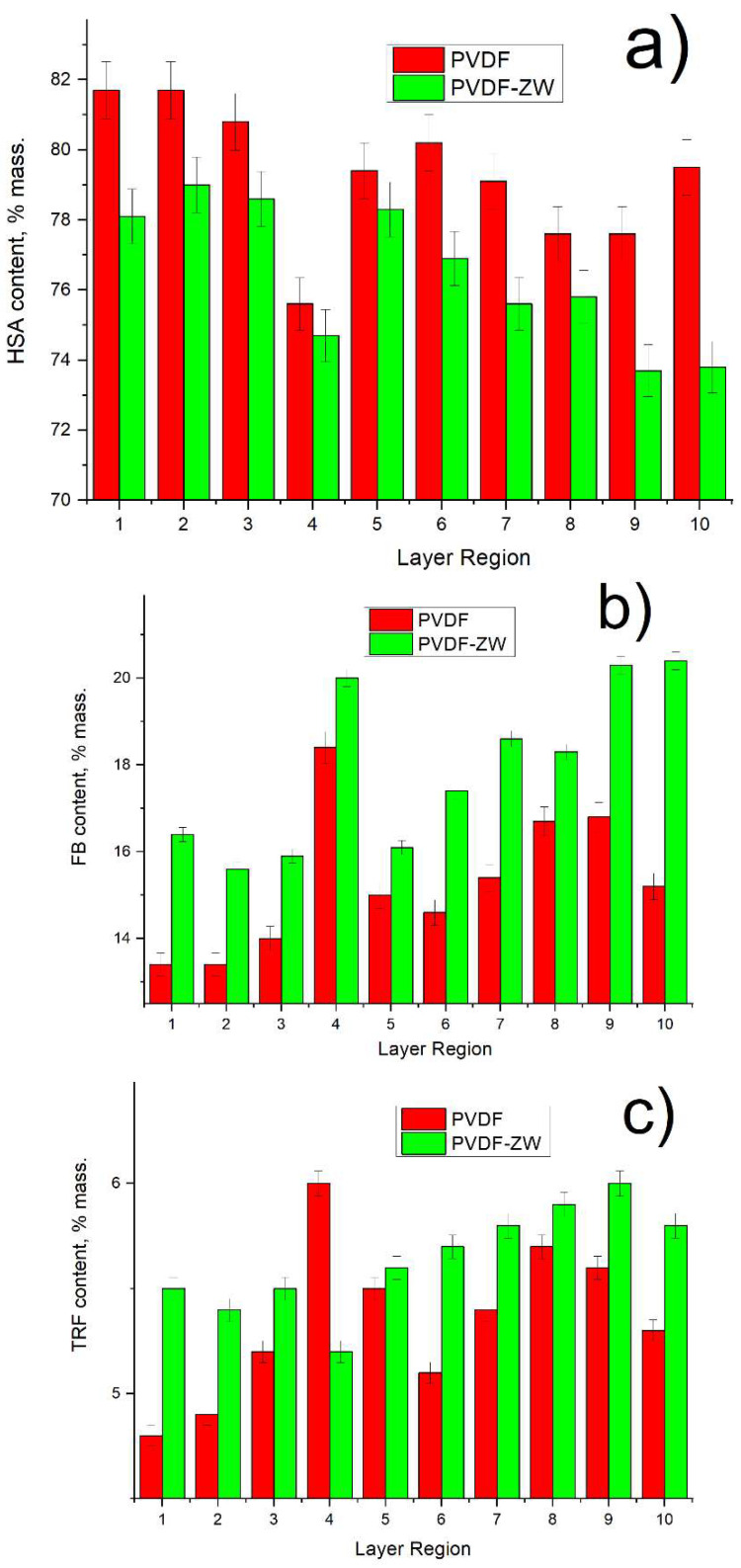
Protein content in adsorbed protein mixture depending on membrane region index: (**a**) HSA; (**b**) FB; (**c**) TRF.

**Figure 7 membranes-13-00117-f007:**
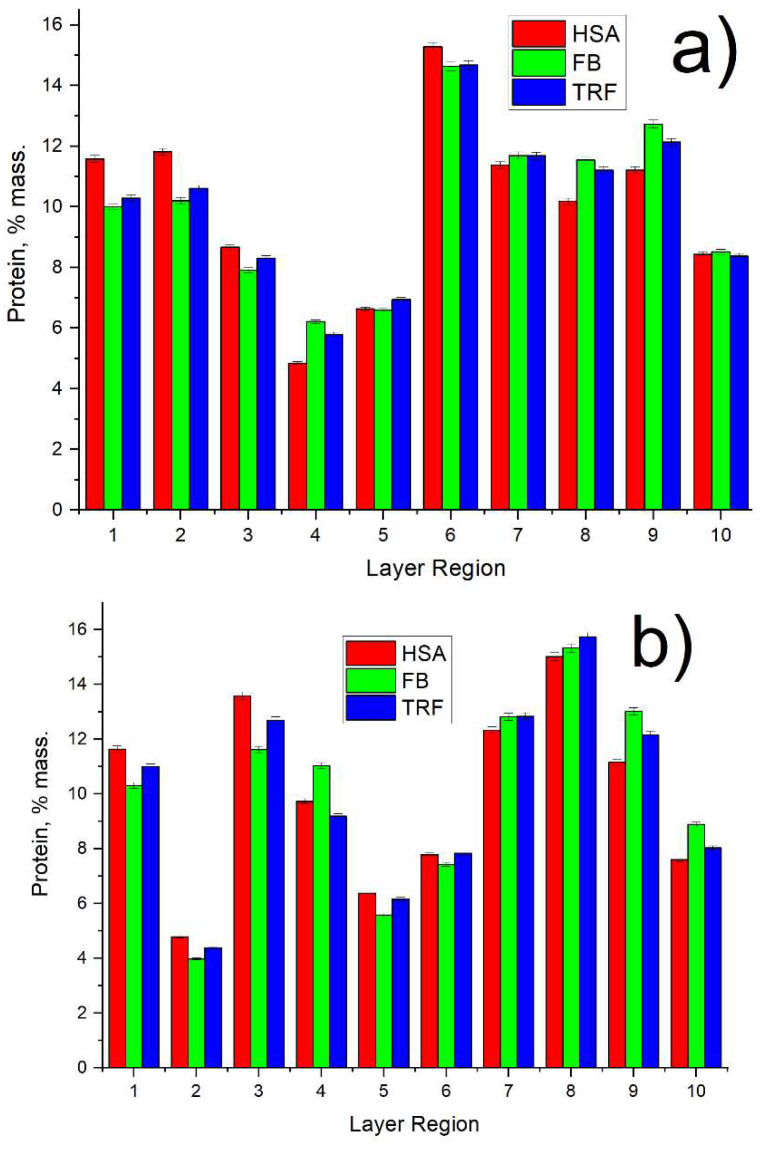
Protein distribution across membrane thickness for: (**a**) PVDF and (**b**) PVDF-ZW membranes.

**Figure 8 membranes-13-00117-f008:**
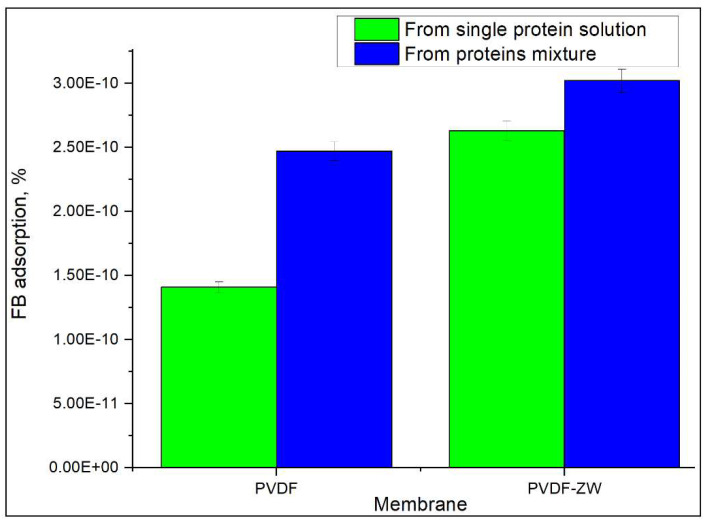
Comparison of FB adsorption from its single solution and protein mixture.

**Figure 9 membranes-13-00117-f009:**
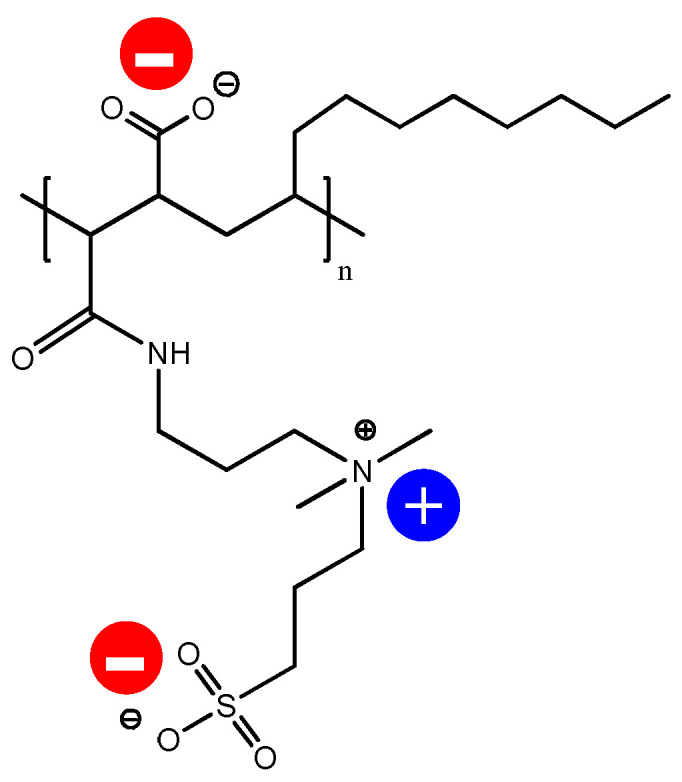
Charges distribution in ZW polymer used for PVDF modification.

**Figure 10 membranes-13-00117-f010:**
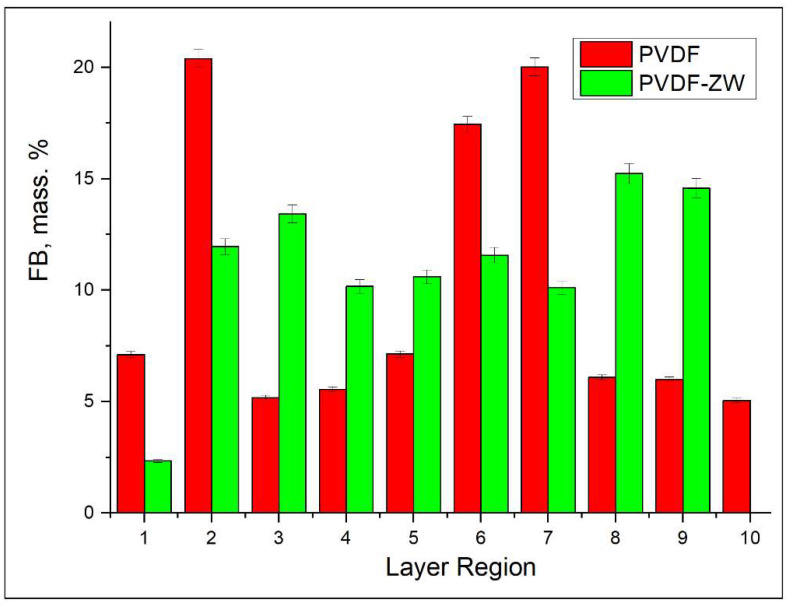
Comparison of FB distribution across membrane (PVDF and PVDF-ZW) thicknesses when adsorbed from a single-protein solution.

## Data Availability

The raw/processed data obtained at the Canadian Light Source (CLS), required to reproduce these findings of this study are available from the corresponding author (Amira Abdelrasoul, A.A.) on reasonable request.

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
