# Peer review of "Investigation on Human Serum Protein Depositions Inside Polyvinylidene Fluoride-Based Dialysis Membrane Layers Using Synchrotron Radiation Micro-Computed Tomography (SR-μCT)"

_membranes, 2023, doi:10.3390/membranes13010117_

Round 1
Reviewer 1 Report
In this manuscript, the authors investigated the human serum protein depositions inside polyvinylidene fluoride-based dialysis membranes layers using synchrotron radiation micro-computed tomography. This work is valuable and interesting. This manuscript could be published in this journal after making minor revisions.
1. Please give the full name for the abbreviations when it first appeared in the paper.
2. The error bars are suggested to be marked in the experimental results.
3. The working mechanism is suggested to be further discussed.
4. In order to highlight the significance of this work, the relevant literatures on the polymer for membrane preparation should be referred to, such as Journal of Colloid and Interface Science, v 630, p 776-785, 2023; Sensors and Actuators B: Chemical, v 289, p 32-41, 2019; Sensors and Actuators B: Chemical, v 255, p 1869-1877, 2018; Sensors and Actuators B: Chemical, v 197, p 66-72, 2014.
Author Response
The authors appreciate the reviewers’ recommendation and comments. Please note that all the scientific/technical pointed out by the reviewer, as comments and queries, within this manuscript have been thoroughly revised. Please, see the point-by-point responses to the reviewer’s comments below; and please, also refer to the applicable pages of the revised manuscript for confirmation. The new changes in this tracked version were highlighted in a “yellow color”. The manuscript was also proofread by a professional editor.
In this manuscript, the authors investigated the human serum protein depositions inside polyvinylidene fluoride-based dialysis membranes layers using synchrotron radiation micro-computed tomography. This work is valuable and interesting. This manuscript could be published in this journal after making minor revisions.
Response: We appreciate the reviewer feedback.
- Please give the full name for the abbreviations when it first appeared in the paper.
Response: All the abbreviations have been clarified as requested
- The error bars are suggested to be marked in the experimental results.
Response: It was added as suggested in all the experimental results (all graphs).
- The working mechanism is suggested to be further discussed.
Response: The discussion has been enhanced and highlighted as suggested.
- In order to highlight the significance of this work, the relevant literatures on the polymer for membrane preparation should be referred to, such as Journal of Colloid and Interface Science, v 630, p 776-785, 2023; Sensors and Actuators B: Chemical, v 289, p 32-41, 2019; Sensors and Actuators B: Chemical, v 255, p 1869-1877, 2018; Sensors and Actuators B: Chemical, v 197, p 66-72, 2014.
Response: Many thanks. It was added as suggested

Reviewer 2 Report
In this study, the authors chemically modified the surface of PVDF membrane using zwitterionic polymer and investigated the effect of the adsorption of human serum proteins on the membrane properties in hemodialysis. Even though this paper reports interesting results, it needs improvement in many parts to be published in “membranes”. My specific opinion is as follows.
1. (L 38) (Apart from the Abstract) The full name of HD (i.e. hemodialysis) should be indicated in the text.
2. (L 70) A detailed review of the references [33-38] should be provided in the text to confirm the originality of this study.
3. (L 87) Please describe the PVDF membrane surface modification method using Zwitterionic polymer in detail.
4. FT-IR spectra (chemical structure) and FE-SEM images (surface and cross-sectional images, morphology) should be provided to verify the successful PVDF membrane surface modification.
5. (All graphs) The title of the Y-axis should be moved to the correct location.
6. (All graphs) Error bars should be marked on the graph to check the reliability of the data.
7. Please add contact angle data.
8. Please clearly explain what the definition of "layer region" is.
9. (L 161) Please explain the "Vroman effect" in detail and indicate references.
10. There is no reference citation in the Results and discussion part. Please indicate references related to interpretation.
11. Please, measure the zeta potentials of the membranes and use them to explain the electrostatic interactions on the membrane surface.
12. Please check the reference notation format and revise it according to the journal's regulations.
Author Response
The authors appreciate the reviewers’ recommendation and comments. Please note that all the scientific/technical pointed out by the reviewer, as comments and queries, within this manuscript have been thoroughly revised. Please, see the point-by-point responses to the reviewer’s comments below; and please, also refer to the applicable pages of the revised manuscript for confirmation. The new changes in this tracked version were highlighted in a “yellow color”. The manuscript was also proofread by a professional editor.
In this study, the authors chemically modified the surface of PVDF membrane using zwitterionic polymer and investigated the effect of the adsorption of human serum proteins on the membrane properties in hemodialysis. Even though this paper reports interesting results, it needs improvement in many parts to be published in “membranes”. My specific opinion is as follows.
Response: We appreciate the reviewer feedback.
- (L 38) (Apart from the Abstract) The full name of HD (i.e. hemodialysis) should be indicated in the text.
Response: It was revised as requested.
- (L 70) A detailed review of the references [33-38] should be provided in the text to confirm the originality of this study.
Response: This section has been revised as requested.
- (L 87) Please describe the PVDF membrane surface modification method using Zwitterionic polymer in detail.
Response: Section 2.2 has been revised and all the details of membrane surface modification has been added.
- FT-IR spectra (chemical structure) and FE-SEM images (surface and cross-sectional images, morphology) should be provided to verify the successful PVDF membrane surface modification.
Response: FT-IR spectra (chemical structure) and FE-SEM data and discussion have been added in Section 3.1 (Figure 3 and Figure 4). Also the experimental methods have been added in Section 2.5.
- (All graphs) The title of the Y-axis should be moved to the correct location.
Response: It was correct in all the graphs.
- (All graphs) Error bars should be marked on the graph to check the reliability of the data.
Response: Error bars have been added as suggested to all the graphs.
- Please add contact angle data.
Response: The available equipment encountered a technical problem however, we have measured the membrane surface charge to reflect the membrane hydrophilicity and its affinity to interact with water molecules and human serum proteins.
- Please clearly explain what the definition of "layer region" is.
Response: A section has been added to the discussion in Section 2.4, in addition to Figure 2.
- (L 161) Please explain the "Vroman effect" in detail and indicate references.
Response: A section has been added to the discussion in Section 3.2.
- There is no reference citation in the Results and discussion part. Please indicate references related to interpretation.
Response: Reference citations in the Results and discussion have been added and highlighted.
- Please, measure the zeta potentials of the membranes and use them to explain the electrostatic interactions on the membrane surface.
Response: The zeta potential measurements have been added and discussed in Section 3.1, in addition the technique has been added in Section 2.5.3.
- Please check the reference notation format and revise it according to the journal's regulations.
Response: References list has been modified as requested.

Round 2
Reviewer 2 Report
The authors have carefully revised the manuscript according to the referees’ comments. In my opinion, this manuscript could be accepted for publication in membranes. However, the resolution of all the graphs should be improved. (L277) [54-55] -> [54,55]